# Geospatial Cellular Distribution of Cancer-Associated Fibroblasts Significantly Impacts Clinical Outcomes in Metastatic Clear Cell Renal Cell Carcinoma

**DOI:** 10.3390/cancers13153743

**Published:** 2021-07-26

**Authors:** Nicholas H. Chakiryan, Gregory J. Kimmel, Youngchul Kim, Joseph O. Johnson, Noel Clark, Ali Hajiran, Andrew Chang, Ahmet M. Aydin, Logan Zemp, Esther Katende, Jad Chahoud, Meghan C. Ferrall-Fairbanks, Philippe E. Spiess, Natasha Francis, Michelle Fournier, Jasreman Dhillon, Jong Y. Park, Liang Wang, James J. Mulé, Philipp M. Altrock, Brandon J. Manley

**Affiliations:** 1Department of Genitourinary Oncology, H. Lee Moffitt Cancer Center and Research Institute, Tampa, FL 33612, USA; Ali.Hajiran@moffitt.org (A.H.); andrew.chang@duke.edu (A.C.); ahmetmurataydin@gmail.com (A.M.A.); logan.zemp@moffitt.org (L.Z.); Esther.Katende@moffitt.org (E.K.); Jad.Chahoud@moffitt.org (J.C.); philippe.spiess@moffitt.org (P.E.S.); Natasha.Francis@moffitt.org (N.F.); michelle.fournier@moffitt.org (M.F.); brandon.manley@moffitt.org (B.J.M.); 2Integrated Mathematical Oncology Department, H. Lee Moffitt Cancer Center and Research Institute, Tampa, FL 33612, USA; gregory.kimmel@moffitt.org (G.J.K.); mferrall.fairbanks@bme.ufl.edu (M.C.F.-F.); philipp.altrock@moffitt.org (P.M.A.); 3Department of Biostatistics and Bioinformatics, H. Lee Moffitt Cancer Center and Research Institute, Tampa, FL 33612, USA; youngchul.kim@moffitt.org; 4Analytic Microcopy Shared Resource, H. Lee Moffitt Cancer Center and Research Institute, Tampa, FL 33612, USA; Joseph.Johnson@moffitt.org; 5Tissue Core Shared Resource, H. Lee Moffitt Cancer Center and Research Institute, Tampa, FL 33612, USA; noel.clark@moffitt.org; 6Department of Pathology, H. Lee Moffitt Cancer Center, Tampa, FL 33612, USA; Jasreman.Dhillon@moffitt.org; 7Department of Cancer Epidemiology, H. Lee Moffitt Cancer Center, Tampa, FL 33612, USA; Jong.Park@moffitt.org; 8Department of Tumor Biology, H. Lee Moffitt Cancer Center and Research Institute, Tampa, FL 33612, USA; Liang.Wang@moffitt.org; 9Immunology Department, H. Lee Moffitt Cancer Center and Research Institute, Tampa, FL 33612, USA; james.mule@moffitt.org

**Keywords:** metastatic clear cell renal cell carcinoma, cancer associated fibroblasts, Ki-67, spatial analysis, immunohistochemistry

## Abstract

**Simple Summary:**

Cancer-associated fibroblasts (CAFs) are highly prevalent cells in the clear cell renal cell carcinoma (ccRCC) tumor immune microenvironment. CAFs are thought to potentiate tumor proliferation primarily through paracrine interactions, as evidenced by laboratory-based studies. We sought to corroborate these findings using surgically removed tissue samples from 96 patients with metastatic ccRCC and associate geospatial relationships between CAFs and rapidly proliferating tumor cells with survival outcomes. We found that CAFs exhibited more geospatial clustering with proliferating tumor cells than with dying tumor cells, and patients whose samples exhibited higher tumor cell proliferation had worse overall survival and were more likely to be resistant to systemic tyrosine-kinase-inhibiting targeted therapies. Immunotherapy resistance was not associated with the geospatial metrics measured in this analysis. Overall, these findings suggest that close proximity to CAFs potentiates tumor cell proliferation, worsening survival and conferring resistance to targeted therapies.

**Abstract:**

Cancer-associated fibroblasts (CAF) are highly prevalent cells in the tumor microenvironment in clear cell renal cell carcinoma (ccRCC). CAFs exhibit a pro-tumor effect in vitro and have been implicated in tumor cell proliferation, metastasis, and treatment resistance. Our objective is to analyze the geospatial distribution of CAFs with proliferating and apoptotic tumor cells in the ccRCC tumor microenvironment and determine associations with survival and systemic treatment. Pre-treatment primary tumor samples were collected from 96 patients with metastatic ccRCC. Three adjacent slices were obtained from 2 tumor-core regions of interest (ROI) per patient, and immunohistochemistry (IHC) staining was performed for αSMA, Ki-67, and caspase-3 to detect CAFs, proliferating cells, and apoptotic cells, respectively. H-scores and cellular density were generated for each marker. ROIs were aligned, and spatial point patterns were generated, which were then used to perform spatial analyses using a normalized Ripley’s K function at a radius of 25 μm (nK(25)). The survival analyses used an optimal cut-point method, maximizing the log-rank statistic, to stratify the IHC-derived metrics into high and low groups. Multivariable Cox regression analyses were performed accounting for age and International Metastatic RCC Database Consortium (IMDC) risk category. Survival outcomes included overall survival (OS) from the date of diagnosis, OS from the date of immunotherapy initiation (OS-IT), and OS from the date of targeted therapy initiation (OS-TT). Therapy resistance was defined as progression-free survival (PFS) <6 months, and therapy response was defined as PFS >9 months. CAFs exhibited higher cellular clustering with Ki-67^+^ cells than with caspase-3^+^ cells (nK(25): Ki-67 1.19; caspase-3 1.05; *p* = 0.04). The median nearest neighbor (NN) distance from CAFs to Ki-67^+^ cells was shorter compared to caspase-3^+^ cells (15 μm vs. 37 μm, respectively; *p* < 0.001). Multivariable Cox regression analyses demonstrated that both high Ki-67^+^ density and H-score were associated with worse OS, OS-IT, and OS-TT. Regarding αSMA+CAFs, only a high H-score was associated with worse OS, OS-IT, and OS-TT. For caspase-3^+^, high H-score and density were associated with worse OS and OS-TT. Patients whose tumors were resistant to targeted therapy (TT) had higher Ki-67 density and H-scores than those who had TT responses. Overall, this ex vivo geospatial analysis of CAF distribution suggests that close proximity clustering of tumor cells and CAFs potentiates tumor cell proliferation, resulting in worse OS and resistance to TT in metastatic ccRCC.

## 1. Introduction

In normal tissue, fibroblast activation and the subsequent release of cytokines, angiogenic mediators, and growth factors are physiologic responses to tissue injury or stress [1,2]. Cancer-associated fibroblasts (CAF) are fibroblasts that have been permanently activated by adjacent tumor cells, which repurpose physiologic fibroblast activity into a pro-tumor survival advantage [1,2,3]. CAFs are highly prevalent cells in the tumor microenvironment in clear cell renal cell carcinoma (ccRCC) and have been implicated in facilitating tumor cell proliferation, angiogenesis, metastasis, and therapy resistance [4,5,6]. Several well-described pro-tumor properties of CAFs are mediated through hypoxia-inducible-factor-1 (HIF1), a pathway that drives oncogenesis in kidney cancer and is upregulated in the majority of ccRCC tumors via alteration of the von-Hippel Lindau (*VHL*) gene [7]. Additionally, molecular receptors mediating the HIF1 pathway are the primary targets for tyrosine-kinase–inhibiting targeted therapies (TT), which are frequently used in metastatic ccRCC [8]. Previous work has demonstrated that increased CAF density in ccRCC tumors is associated with worse overall survival (OS) [4,6].

The induction of tumor cell proliferation has been postulated as a major mechanism of CAF-mediated pro-tumor activity [1]. Immunohistochemical (IHC) staining with Ki-67, a nuclear protein that is present during active phases of the cell cycle and absent from resting cells, has been shown to be an excellent marker for identifying rapidly proliferating tumor cells [9,10]. This staining is clinically relevant for grading and prognosis in several primary cancer sites, including breast and colorectal cancers [11,12,13]. Currently, there is no role for Ki-67 staining in ccRCC guideline-based clinical practice, though a handful of studies have associated high Ki-67 staining with more advanced disease stage, worse OS, and worse cancer-specific survival (CSS) [14,15,16,17].

Despite the well-described in vitro relationship between CAFs and increased tumor cell proliferation, no prior study has investigated the interplay between CAFs and rapidly proliferating tumor cells in ccRCC tumor samples. The discovery of distinct infiltration patterns and spatial relationships between CAFs and rapidly proliferating tumor cells could support previous in vitro study findings with ex vivo evidence and inform ongoing research into microenvironment-modulating antineoplastic therapies targeting fibroblasts [18]. Our primary objective was to investigate cellular distribution patterns and spatial relationships between CAFs and proliferating and apoptotic tumor cells in primary tumor samples from patients with metastatic ccRCC. Additionally, to evaluate the associations between these measures and OS. Our secondary objective was to determine associations between treatment outcomes and targeted therapy and immunotherapy, as well as define spatial relationships between CAFs, proliferating tumors cells, and apoptotic tumor cells.

## 2. Materials and Methods

### 2.1. Patient and Sample Selection

Samples were included from patients with primary ccRCC who had metastatic ccRCC at the time of sample collection and whose tumor specimens were available in formalin-fixed paraffin-embedded blocks. Included patients had received either TT, immunotherapy (IT), or combination TT/IT as systemic treatment from October 2004 to September 2020. Written informed consent was obtained from all tissue donors. All tumor and normal tissue samples were obtained through protocols approved by the institutional review board (H. Lee Moffitt Cancer Center and Research Institute’s Total Cancer Care protocol MCC# 14690; Advarra IRB Pro00014441). The general workflow of our methods is shown in Figure 1.

### 2.2. Immunohistochemical Specimen Preparation

Three immediately adjacent slides (3 µm thickness) were prepared from each tissue block. IHC staining for alpha-smooth muscle actin ((αSMA a marker for activated fibroblasts), Ki-67 (a marker for proliferating cells), and caspase-3 (a marker for cells undergoing apoptosis) were performed; 1 stain was used on each slide. Slides were stained using a Ventana Discovery XT automated system (Ventana Medical Systems, Tucson, AZ, USA) as per manufacturer’s protocol, with proprietary reagents. Briefly, slides were deparaffinized on the automated system with the EZ Prep solution (Ventana). A heat-induced antigen retrieval method was used in Cell Conditioning (Ventana). The rabbit primary antibody that reacts to Ki-67 (#790-4286 (Ventana)) was used at a prediluted strength and incubated for 16 min. The rabbit primary antibody that reacts to Cleaved Caspase 3 (#9661 (Cell Signaling, Danvers, MA, USA)) was used at a 1:4000 concentration in Dako antibody diluent (Carpenteria, CA, USA) and incubated for 60 min. The rabbit primary antibody that reacts to αSMA (#ab32575 (Abcam, Cambridge, MA, USA)) was used at a 1:250 concentration in Dako antibody diluent (Carpenteria, CA, USA) and incubated for 32 min. For all stains, the Ventana OmniMap Anti-Rabbit Secondary Antibody was used for 16 min. The detection system used was the Ventana ChromoMap kit, and slides were then counterstained with hematoxylin. Slides were then dehydrated and cover-slipped as per normal laboratory protocol.

### 2.3. Quantitative Digital Image Analysis

Slides were digitized with a Leica Aperio AT2 slide scanner (Vista, CA, USA) using a 20 X/0.75 NA objective lens. An experienced genitourinary pathologist (JD) used the annotation pen tool in the Aperio Imagescope software to define the tumor-core zone in each hematoxylin and eosin image. High-resolution Aperio SVS images representing the 3 IHC stains were imported into Visiopharm (Hoersholm, Denmark) for quantitative digital image analysis. We leveraged a tissue alignment algorithm available in Visiopharm to optimally align 3 adjacent slides of each patient sample set. These slides were visually inspected to ensure appropriate alignment. Two equally sized regions of interest (ROI) were selected from the tumor-core zone from each image; tumor cells were evenly distributed throughout each ROI, and cellular appearance was consistent with that seen in the remainder of the slide. The ROI sizes were standardized at 3426 × 1379 pixels, at a pixel resolution of 0.502 µm/pixel. Thresholds for staining positivity were set by an experienced digital pathology image analyst (JJ) and confirmed by a study pathologist (JD). These thresholds were used with Visiopharm’s cell detection algorithms to identify and categorize cells into negative, weak, moderate, and strong bins on the basis of staining intensity. This intensity-based distribution of cells is comparable to the qualitative method used by pathologists, in which staining intensity is categorized as 0, 1+, 2+, and 3+ [19]. Percent positivity and H-Scores were calculated for each ROI using this intensity data. H-score is an approach that globally quantifies intensity and percent positivity throughout the entire ROI into 1 score, according to the following formula: H-score = (1 × (%cells weak)) + (2 × (%cells moderate)) + (3 × (%cells strong)) [20]. H-scores range from 0 to 300, with 0 representing no cell staining for the marker of interest and 300 representing every cell staining with the highest intensity [20]. Additionally, Cartesian coordinates for the (x,y) location of each cell’s central mass, with each cell’s associated marker status, were abstracted from the digital image using Visiopharm.

### 2.4. Cellular Distribution and Spatial Analysis

For each ROI, the per-cell Cartesian coordinate and marker positivity data were converted into spatial point patterns. Cell density was calculated as the number of positive cells per mm^2^. ROIs containing ≥10 cells positive for a relevant marker were considered eligible for spatial analysis. As there is no previously validated standard for this cutoff, the ≥10 cell cutoff was agreed upon through the consensus of the authors. Cells classified as exhibiting strong staining intensity, as defined above, were considered positive for the purpose of spatial analysis. Cellular clustering was quantified using Ripley’s K function, a methodology for quantifying spatial heterogeneity that is most commonly used in ecology and economics, with isotropic edge correction, and the following normalization was applied: *nK*(*r*) = *K*(*r*)/*πr*^2^, as described previously [21,22]. As such, the expected value of nK(r) for complete spatial randomness is 1.0, assuming a homogenous Poisson process [21]. Values of nK(r) > 1.0 represent cellular clustering and values < 1.0 represent cellular dispersion. The range of possible values for nK(r) is 0 to infinity. The nK(r) value is an observed over expected ratio (i.e., αSMA/Ki-67 nK(25 µm) = 1.30 can be interpreted as: “There were 30% more Ki-67^+^ cells within a 25 µm radius of each αSMA cell than would be expected if the cells were randomly distributed.”).

Initially, two search-circle radii were utilized for the spatial analysis. To reflect cellular clustering at a localized distance, nK(r) at a radius of 25 µm was used in this analysis and will henceforth be referred to as nK(25). The search-circle radius value of 25 µm was selected, as it represents approximately double that of a typical ccRCC tumor cell radius and, as such, should represent the area in the immediate vicinity of the cell. To reflect a more global view of cellular clustering, a radius of 125 µm was utilized, which will be referenced as nK(125). Subsequently, we determined that Spearman’s correlations demonstrated a very strong correlation between nK(25) and nK(125) (r > 0.90). Given this redundancy, we utilized nK(25), not nK(125), for the remainder of the analysis.

As a second independent measure of spatial analysis, linear nearest neighbor distances were determined from each cell to its nearest neighbor cell among each of the 3 cell types. Scaled two-dimensional kernel density plots were generated to analyze the distribution of nearest neighbor distances from each αSMA^+^ cell to its nearest neighbor caspase-3^+^ and Ki-67^+^ cell. These nearest neighbor density plots were generated with the cohort stratified by ccRCC versus normal kidney, treatment response versus resistance to TT, and treatment response versus resistance to IT.

### 2.5. Variable Definitions

Age was defined as life years at the time of ccRCC diagnosis. Tumor grade was defined per the histologic classification criteria proposed by the International Society of Urologic Pathologists and implemented by the World Health Organization (ISUP/WHO) [23]. International Metastatic RCC Database (IMDC) scores were determined for each patient and categorized into good, intermediate, and poor risk groups, as previously described [24]. OS was determined from the date of metastatic ccRCC diagnosis to the date of death or censoring at the last follow-up. Treatments classified as IT included both immune checkpoint inhibitors and high-dose interleukin-2 (IL-2). Treatments classified as TT included small-molecule tyrosine-kinase-inhibiting therapies approved for first-line treatment in ccRCC [17]. OS from the date of the first receipt of IT (OS-IT) and OS from the date of first receipt of TT (OS-TT) were also determined. Response to therapy was defined as clinical progression-free survival greater than 9 months from treatment initiation, and resistance was defined as progression <6 months from treatment initiation. Patients who progressed between 6 and 9 months after initiating therapy were not classified as responsive or resistant. Statistical significance was defined as a two-tailed alpha-risk of 0.05 or less.

### 2.6. Statistical Analysis

In addition to demographic, clinical, and pathologic characteristics, patient-level data included marker densities and intensities, H-scores, and uni- and bi-variate spatial distribution metrics for each of the 3 included IHC markers. As each patient had 2 ROIs analyzed, the IHC-derived metrics were averaged such that each patient had 1 value for each metric. Spearman’s correlation coefficients were determined between each pairwise combination to assess the interactions between the IHC-derived metrics.

As standardized cutoffs do not exist for the IHC-derived metrics utilized in this analysis, optimal cut-points were determined for each metric, maximizing the log-rank test statistics for OS, as previously described [25]. The survival analysis used a multivariable Cox proportional hazards regression for each metric, utilizing age and IMDC risk category as covariates. False discovery rate- (FDR-)adjusted *p* values were calculated for multiple comparison correction. This analysis was repeated for the OS-IT and OS-TT endpoints.

Patients were then stratified by response or non-response to IT and TT, and values of each IHC-derived metric were compared between responders and non-responders using a two-sample t-test.

The distribution of nearest neighbor distances from αSMA^+^ cells to their nearest caspase-3^+^ cell was compared to that of the distance to the nearest Ki-67^-^ cell using a Wilcoxon test. This process was repeated after stratifying the cohort by IT response and non-response and TT response and non-response.

### 2.7. Statistical Software

All statistical and spatial analyses were performed using R version 4.0.2 (The R Foundation for Statistical Computing (Vienna, Austria)). The *maxstat* package was used to determine optimal cut-points, *survival* and *survminer* packages used for the survival analysis, and the *spatstat*, *MASS*, and *FNN* packages for spatial analysis.

## 3. Results

### 3.1. Study Population

The study population included 96 patients (median age, 60 years (IQR 33–87); male, 68 (71%); white race, 90 (94%)) (Table 1). Fifty-one patients (53%) underwent first-line IT (18 responders, 25 resistant, and 8 indeterminate); 42 patients (44%) had first-line TT (27 responders, 9 resistant, and 6 indeterminate); and 3 patients (3%) had first-line combination IT/TT. Median follow-up was 54 months (IQR, 41–97) for the 37 patients (38%) who were alive at last contact.

### 3.2. CAFs Are Highly Prevalent in the ccRCC Tumor Microenvironment

αSMA^+^ CAFs had the highest median cell density and H-score (density, 1904 cells/mm^2^ (IQR 1266–3382); H-score 35.5 (IQR 24.0–51.2)), as compared with respective values for Ki-67^+^ and caspase-3^+^ tumor cells (Ki-67^+^ density, 1087 cells/mm^2^ (IQR, 433–1909); Ki-67 H-score 7.0 (IQR 3.1–15.5); caspase-3^+^ density, 203 cells/mm^2^ (IQR 64–547); caspase-3^+^ H-score, 0.8 (IQR, 0.3–2.2)) (Wilcoxon *p*  <  0.001 for all comparisons within density and H-score groups) (Figure 2A).

### 3.3. CAFs Are Spatially Clustered with Proliferating Tumor Cells

Using the normalized Ripley’s K function, we found that αSMA^+^ CAFs were more spatially clustered with Ki-67^+^ tumor cells than they were with caspase-3^+^ tumor cells (nK(25), 1.15 vs. 1.09, respectively; *p* = 0.045) (Figure 2B). Caspase-3^+^ tumor cells were more spatially clustered with Ki-67^+^ tumor cells than they were with αSMA^+^ CAFs (nK(25), 1.16 vs. 1.09, respectively; *p* = 0.035). The amount of cellular clustering demonstrated between αSMA^+^ CAFs and Ki-67^+^ cells was similar to that between Ki-67^+^ tumor cells and caspase-3^+^ tumor cells (*p* = 0.434).

The median nearest-neighbor (NN) distance from αSMA^+^ CAFs to the nearest Ki-67^+^ cell was shorter than the distance to the nearest caspase-3^+^ cell (14.4 vs. 34.9 µm; *p* < 0.001) (Figure 2C). The median NN distance from Ki-67^+^ cells to the nearest caspase-3^+^ cell was shorter than the distance from αSMA^+^ CAFs to the nearest caspase-3^+^ cell (31.3 vs. 34.9 µm; *p* < 0.001).

Overall, the density and spatial analyses suggest a tumor architecture in which αSMA^+^ CAFs and Ki-67^+^ tumor cells demonstrate the highest clustering and shortest NN distance, αSMA^+^ CAFs and caspase-3^+^ tumor cells have the least amount of clustering and longest NN distance, and cell densities are the highest for αSMA^+^ CAFs and lowest for caspase-3^+^ tumor cells. Figure 3 shows an example of the hypothetical median tumor architecture of these 3 markers, as suggested by the distribution and spatial analysis metrics for the entire cohort.

### 3.4. CAF Density and αSMA H-Score Are Positively Correlated with Proliferating and Apoptotic Tumor Cells

Spearman’s correlations demonstrated that the αSMA H-score had a moderate positive correlation (r = 0.4–0.6) with αSMA density, Ki-67 H-score, and Caspase H-score and a weak positive correlation (r = 0.2–0.4) with caspase-3 density and Ki-67 density (Figure 2D). αSMA density had a moderate positive correlation with Ki-67 density, caspase-3 density, and αSMA H-sore and a weak positive correlation with Ki-67 H-score and caspase-3 H-score. Clustering metrics and density/H-score metrics were consistently found to have either no correlation or a weak negative correlation (r = −0.4–0). Clustering metrics at a local or global scale (nK(25) or nK(125), respectively) were found to have a strong positive correlation with each other within marker-types (r = 0.6–1.0). Complete information regarding pairwise correlations between the IHC-derived metrics can be found in Figure 2D.

None of the IHC-derived metrics were significantly correlated with primary tumor size or grade, confirming that these measures are not simply surrogates for conventionally measures (Appendix A).

### 3.5. High αSMA, Ki-67, and Caspase-3 H-Scores Are Associated with Significantly Worse OS

Multivariable Cox proportional hazards regression analysis showed that patients with a high αSMA H-core had significantly worse OS (HR 1.45 (1.05–2.01); *p* = 0.02), OS-IT (HR 1.51 (1.05–2.17); *p* = 0.03), and OS-TT (HR 1.46 (1.06–2.02); *p* = 0.012 than patients with a low αSMA H-score. Interestingly, stratification by αSMA CAF cell density alone had no statistically significant association with OS, OS-IT, or OS-TT (*p* = 0.32, 0.17, and 0.24, respectively) compared with patients with low αSMA^+^ CAF cell density (Figure 4A).

Patients with a high Ki-67 H-score had significantly worse OS (HR 1.79 (1.40–2.29); *p* < 0.001), OS-IT (HR 1.47 (1.13–1.91); *p* < 0.001), and OS-TT (HR 1.71 (1.35–2.17); *p* < 0.001) than patients with a low Ki-67 H-score. Patients with high Ki-67^+^ cell density had significantly worse OS (HR, 1.30 (1.06–1.61); *p* = 0.01), and OS-TT (HR, 1.27 (1.04–1.57); *p* = 0.02), with no significant difference in OS-IT (HR, 1.21 (0.98–1.50); *p* = 0.08), than patients with low Ki-67^+^ cell density (Figure 4B).

Patients with a high caspase-3 H-score had significantly worse OS (HR 1.33 (1.01–1.75); *p* = 0.04) and OS-TT (HR 1.46 (1.14–1.88); *p* < 0.001), with no significant difference in OS-IT (HR 1.19 (0.89–1.58); *p* = 0.24), than patients with low caspase-3 H-scores. Patients with high caspase-3^+^ cell density had significantly worse OS (HR 1.26 (1.02–1.56); *p* = 0.03), and OS-TT (HR 1.25 (1.02–1.53); *p* = 0.03), with no significant difference in OS-IT (HR 1.18 (0.95–1.46); *p* = 0.13) (Figure 4C).

Using Ripley’s K function, we did not find any spatial clustering metrics predictive of OS, OS-IT, or OS-TT. The results of the multivariable Cox regression analysis assessing all IHC-derived H-scores, densities, and spatial clustering metrics are available in Table 2.

### 3.6. High Ki-67 Density and H-Score Are Associated with Resistance to First-Line TT

To examine specific outcomes to first-line therapy, we stratified patients by response and resistance to TT or IT. Patients who were resistant to TT had higher Ki-67 H-scores and higher Ki-67^+^ cell densities (Wilcoxon *p* values = 0.013 and 0.035, respectively; Figure 5). None of the other IHC-derived metrics were associated with TT or IT response or resistance in the first-line setting.

### 3.7. Distinct Geospatial Distributions of CAFs Are Associated with Treatment Response

Examining our cellular marker distributions, we found ccRCC and normal kidney samples to have dramatically different kernel density distributions with respect to their NN distances from each marked cell type (Figure 5A(i,ii)). ccRCC samples had a shorter NN distance from αSMA^+^ to Ki-67^+^ cells, compared with normal kidney samples (13 vs. 31 µm, respectively; *p* < 0.001), and longer NN distance from αSMA^+^ to caspase-3^+^ cells (32 vs. 9 µm, respectively; *p* < 0.001) (Figure 6A(iii)).

TT-responsive and TT-resistant patient samples had significantly different kernel density distributions with respect to their NN distances from each marked cell type (Figure 5B(ii) and Figure 6B(i)). TT-responsive patient samples had a longer NN distance from αSMA^+^ to caspase-3^+^ cells (31 vs. 21 µm, respectively; *p* < 0.001) (Figure 6B(iii)).

IT-responsive and IT-resistant patient samples had significantly different kernel density distributions with respect to their NN distances from each marked cell type (Figure 6C(i,ii)). IT-responsive patient samples had shorter NN distance from αSMA^+^ to caspase-3^+^ cells (24 vs. 35 µm, respectively; *p* < 0.001) (Figure 6C(iii)).

## 4. Discussion

Our geospatial analysis of CAF distribution demonstrated that close-proximity clustering of CAFs and tumor cells might potentiate tumor cell proliferation. It also shows that a higher density and staining intensity of CAFs and proliferating tumor cells is associated with poor OS and systemic treatment outcomes. To our knowledge, this is the first ex vivo analysis associating CAFs with the potentiation of tumor cell proliferation in metastatic ccRCC.

We found that CAFs were significantly more clustered with proliferating than apoptotic tumor cells. Overall, these findings, as illustrated in Figure 3, show that close tumor cell proximity to CAFs potentiates proliferation. Additionally, this spatial architecture suggests that tumor cells may be in competition with one another for resources provided by CAFs, with the tumor cells further away from CAFs succumbing to apoptosis; however, this analysis was not designed to directly test this hypothesis.

Though this is the first study to directly measure ex vivo clustering of CAFs with proliferating tumor cells in ccRCC, prior studies have demonstrated this effect in vitro, and several ex vivo analyses have discovered pathways that presumably rely on close proximity [1,2,3,7,18]. Notably, CAFs have been found to undergo a metabolic shift when exposed to adjacent cancer cells. This shift results in a Warburg-like glycolytic metabolism in CAFs supplying lactate for cancer cells to fuel the Krebs cycle, which leads to anabolic growth and tumor cell proliferation [7]. Presumably, this is a paracrine effect, with CAF-adjacent tumor cells competing with one another for lactate. This metabolic shift is impacted via the HIF1 pathway, which is of particular relevance in ccRCC, a disease in which the vast majority of tumors harbor a somatic alteration in the *VHL* gene. The *VHL* alteration results in dramatically increased intracellular HIFα, which would be expected to further enhance the aforementioned metabolic shift [26]. This well-described effect of CAFs on the tumor microenvironment lends significant biologic plausibility to the geospatial findings identified in this analysis.

Similarly, hypoxia in the tumor microenvironment has been identified as a significant determinant of the extracellular matrix composition in tumors, resulting in increased *HIF1A* expression. This subsequently increases the production of growth factors that can potentiate tumor cell proliferation and trigger fibroblast activation and fibrosis [27]. In the >80% of ccRCC tumors that harbor somatic *VHL* mutations, the resulting derangement of HIFα metabolism would be expected to result in a tumor microenvironment consistent with extreme hypoxic conditions. This resultant state would be in place regardless of the true state of oxygen availability within the tissue [26]. This is another example of significant overlap existing between ccRCC and CAF-potentiated molecular pathways.

Additionally, this analysis of a metastatic ccRCC cohort demonstrated strong associations between increased CAF staining intensity and inferior OS while accounting for age and IMDC risk category. This finding corroborates previous IHC studies of ccRCC patients that identified an association between increasing CAF density and OS, as well as a more advanced stage at diagnosis. However, these cohorts did not include stage IV patients [4,5,6].

While using Ki-67 staining, this analysis also identified worse OS associated with increased density of proliferating tumor cells. Though not yet adopted into clinical practice, increased Ki-67 staining has been shown in several studies to be associated with worse survival in ccRCC patients across a wide range of clinical stages [14,15,16,28]. Cell cycle proliferation (CCP) scoring, an RNA-seq-based gene signature score quantifying the expression level of several genes associated with tumor cell proliferation, has been recently described as a promising prognostic biomarker for predicting poor survival and adverse pathology in patients with clinically localized ccRCC [29,30]. Ki-67 IHC staining is a simpler and less costly method than the CCP score and may yield similar biologic information. Further studies are warranted to determine agreement and concordance between Ki-67 IHC staining and CCP score.

In addition to survival metrics, we identified several promising associations between geospatial distributions and treatment response and resistance. Patients who were resistant to TT had increased Ki-67 density and H-scores, which to our knowledge, has not been previously reported. Additionally, two-dimensional kernel-density plots depicting the NN distances of CAFs to their nearest Ki-67^+^ and caspase-3^+^ cellular neighbors yielded density distributions that are clearly disparate between TT- and IT-responding and resistant tumors. The difference between these distributions was primarily driven by the distance from CAFs to their nearest caspase-3^+^ neighbor, with TT responders having significantly longer CAF-caspase NN distance than patients with TT resistance and IT responders having significantly shorter CAF-caspase NN distance than patients with IT resistance. These novel findings suggest the possibility of distinct geospatial CAF architectures associated with response to TT versus IT.

There are several limitations of our study that deserve mentioning. First, the patient cohort studied was heterogeneous and included a variety of systemic treatment agents and was determined partially on the availability of tumor samples. This limited the applicability of our results with contemporary systematic treatment regimens. Secondly, we used a limited number of cellular markers, which likely oversimplified the underlying biology of the tumor microenvironment. There are limits to the practicality of examining an exhaustive list of cellular markers in a novel study such as this and interpreting their possible clinical impact. However, this simplification did allow us to examine these specific markers in a robust fashion and identify possible spatial interactions deserving of future investigations. Additionally, αSMA staining is not 100% specific for CAFs but is a marker for activated fibroblasts that can also stain pericytes and smooth muscle cells. Additionally, CAFs are a heterogeneous group of cells with several distinct subtypes. This study was not designed to assess subclassifications of CAFs. Additionally, non-malignant cells such as infiltrating immune cells can also stain positive for Ki-67, potentially resulting in false-positive staining in some cases. Lastly, we limited our study to using 2 of the more commonly employed spatial metrics, Ripley’s K and NN distance, while acknowledging that many other spatial metrics have been defined in the literature.

## 5. Conclusions

This ex vivo geospatial analysis of CAF distribution in ccRCC samples suggests that close-proximity clustering of tumor cells and αSMA+CAFs is associated with tumor cell proliferation. Increased tumor cell proliferation was associated with worse OS and resistance to TT. Patients with high αSMA+CAF density and tumor cell proliferation had significantly worse OS from the time of immunotherapy initiation.

## Figures and Tables

**Figure 1 cancers-13-03743-f001:**
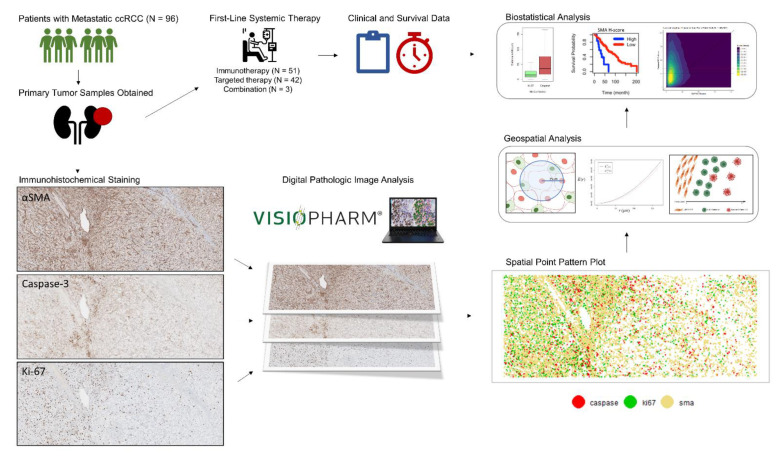
General project workflow for the analysis.

**Figure 2 cancers-13-03743-f002:**
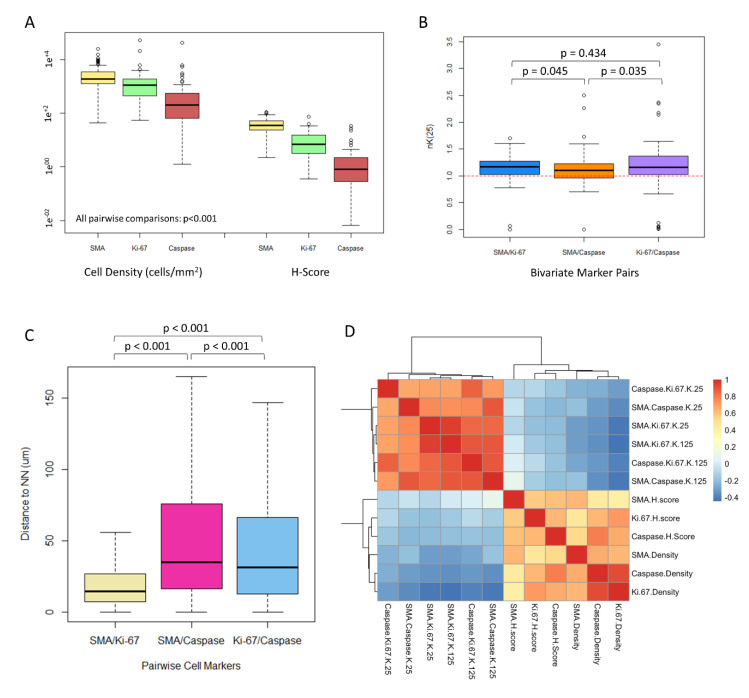
Geospatial metrics quantifying cell-cell relationships. (**A**): Boxplot diagrams depicting median cell density (cells/mm^2^) and H-scores for each of the immunohistochemistry stains used in the analysis (αSMA, Ki-67, and caspase-3). Wilcoxon *p* values < 0.001 for all comparisons. (**B**): Boxplot diagrams depicting bivariate spatial clustering, nK(25) and values for each pairwise combination of cells for patient-level data (*n* = 96 patients). Values > 1.0 reflect clustering of the paired cells, and values < 1.0 reflect dispersion. Wilcoxon *p* values displayed in the plot. (**C**): Boxplots depicting median nearest neighbor distances in µm from the indicated pairwise cell combinations for cellular-level data (*n* > 500,000 cells). Wilcoxon *p* values are displayed in the plot. (**D**): Heatmap diagram depicting Spearman’s correlation coefficients between each immunohistochemistry-derived measurement used in the analysis. Note that the deepest blue values reflect a Spearman’s correlation coefficient of −0.4 and the deepest red1.0.

**Figure 3 cancers-13-03743-f003:**
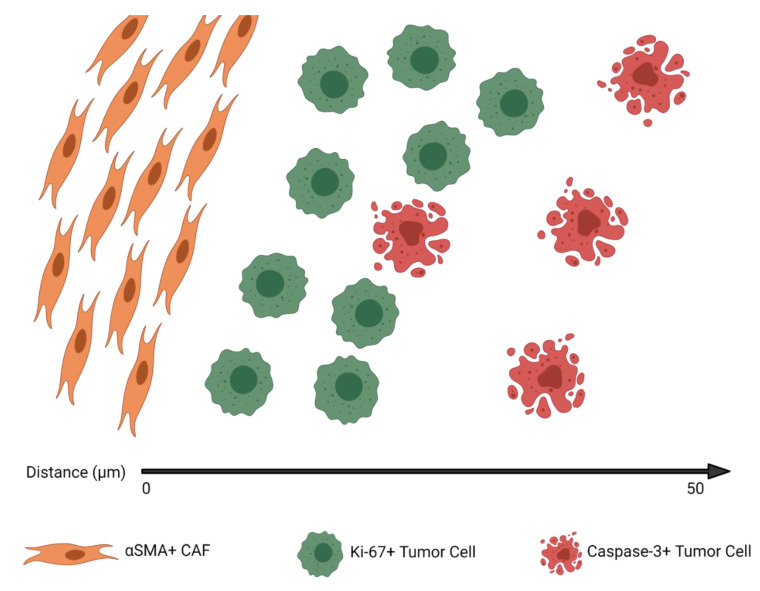
The median tumor architecture identified on immunohistochemistry, derived from the marked cell density and clustering metrics described in Figure 2.

**Figure 4 cancers-13-03743-f004:**
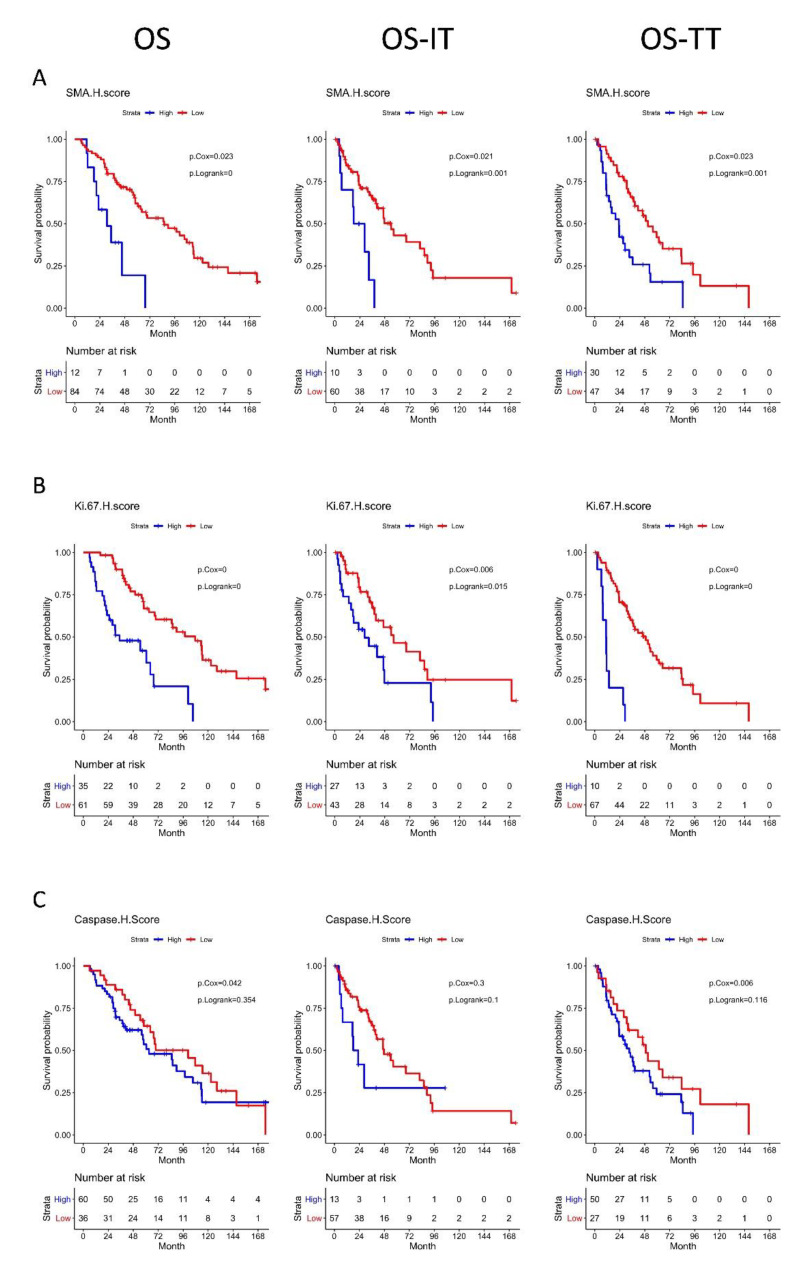
Kaplan–Meier curves for overall survival, overall survival for the date of immunotherapy initiation, and overall survival from the date of targeted therapy initiation, with groups stratified by (**A**): αSMA H-score, (**B**): Ki-67 H-score, and (**C**): Caspase H-score. Multivariable Cox regression and log-rank *p* values were reported within the plots.

**Figure 5 cancers-13-03743-f005:**
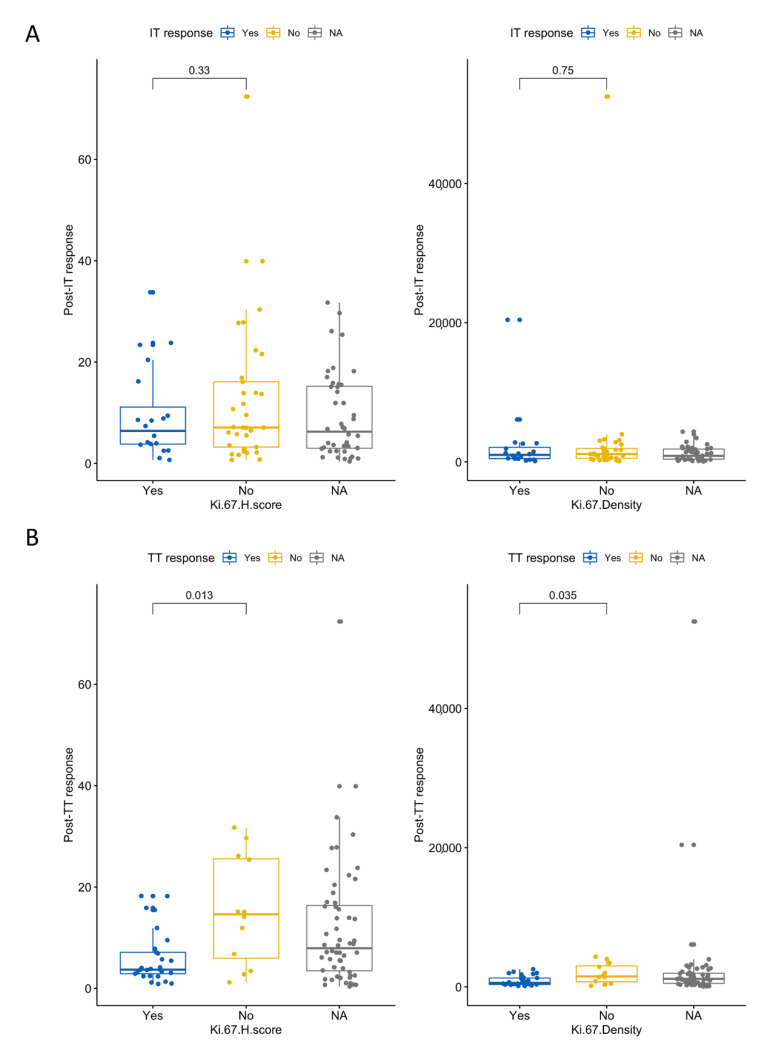
Ki-67 H-scores and densities stratified by response to first-line therapy with (**A**): immunotherapy and (**B**): targeted therapy. The NA category represents patients that did not receive that respective therapy as a first-line agent.

**Figure 6 cancers-13-03743-f006:**
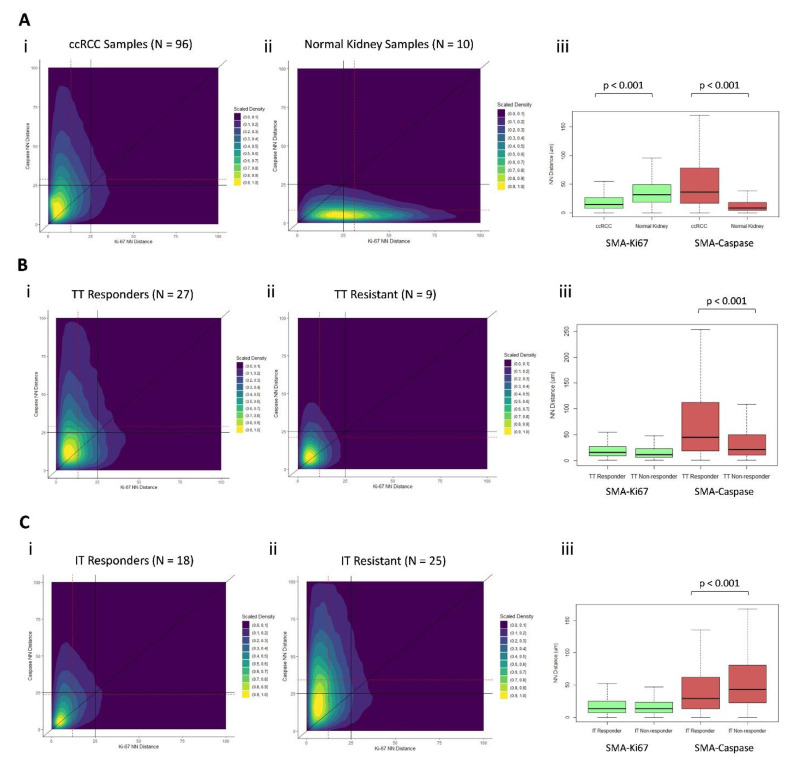
Scaled kernel density plots (i and ii’s), with the x-axis representing the distance from each αSMA^+^ cell to its nearest Ki-67^+^ neighbor (µm) and the y-axis representing the distance from each αSMA^+^ cell to its nearest caspase-3^+^ neighbor (µm). Each point on the plot represents an individual αSMA^+^ cell, and the location of the point corresponds with cells’ NN distance from the nearest Ki-67^+^ and caspase-3^+^ neighbor. Two-dimensional kernel density is graphically depicted, overlying the scatter plot, and has been scaled to a range of 0–1.0 in each plot. Dotted red lines reflect median values for the x and y axes. Solid black lines are anchored at fixed values to facilitate visual interpretation between plots. Boxplots (iii’s) compare median nearest-neighbor (NN) distances (µm) from αSMA^+^ to NN Ki-67^+^ and αSMA^+^ to NN caspase-3^+^ cells, stratified by groups as indicated. (**Ai**) All ccRCC samples (*n* = 96), (**Aii**) Normal kidney samples (*n* = 10). (**Bi**) Targeted therapy responders (*n* = 27). (**Bii**) Targeted therapy-resistant (*n* = 9). (**Ci**) Immunotherapy responders (*n* = 18). (**Cii**)-Immunotherapy-resistant (*n* = 25). (**Aiii**, **Biii** and **Ciii**) Boxplots representing the nearest neighbor distances for each pairwise cell-cell distance for (**Aiii**) ccRCC verus normal, (**Biii**) Targeted therapy response versus non-response, and (**Ciii**) Immunotherapy response versus non-response.

**Table 1 cancers-13-03743-t001:** Baseline patient, tumor, and treatment characteristics, *n* = 96 ^1.^

Characteristic.	No. (%)
Median age, range	59 (55–67)
Gender	
Female	28 (29)
Male	68 (71)
Race	
White	90 (94)
Black	2 (2.1)
Asian	1 (1.0)
Other	3 (3.1)
IMDC Risk Category	
Favorable	20 (21)
Intermediate	56 (58)
Poor	9 (9)
Indeterminate	11 (12)
Primary tumor size (cm)	8.2 (6.0, 11.0)
ISUP Grade	
2	11 (11)
3	60 (62)
4	25 (26)
Sarcomatoid Variant	
No	87 (91)
Yes	9 (9.4)
Rhabdoid Variant	
No	88 (92)
Yes	8 (8.3)
First Line Therapy	
IT	51 (53)
TT	42 (44)
Both	3 (3.1)
Response Category	
IT Resistance	25 (32)
IT Response	18 (23)
IT Indeterminate	8 (8.3)
TT Resistance	9 (9.4)
TT Response	27 (28)
TT Indeterminate	6 (6.3)

^1^ Statistics presented: median (IQR); Abbreviations: IMDC, International Metastatic RCC Database Consortium; ISUP, International Society of Urologic Pathologists; IT, immunotherapy; TT, targeted therapy.

**Table 2 cancers-13-03743-t002:** Multivariable Cox regressions using the covariates age and International Metastatic RCC Database Consortium risk category for the outcomes of (**A.**) overall survival, (**B.**) overall survival from the date of immunotherapy initiation, and (**C.**) overall survival from the date of targeted therapy initiation.

**A.**								
**Metric**	**Cutoff**	**N High**	**N Low**	**Survival High (mo.)**	**Survival Low (mo.)**	**Cox HR**	**95% CI**	***p***
SMA H-score	37.028	12	84	63.2	97.6	1.45	1.05–2.01	0.02
Ki 67 H-score	7.088	35	61	54.7	97.6	1.79	1.40–2.29	<0.001
Caspase H-Score	0.834	60	36	85.8	68.6	1.33	1.01–1.75	0.04
SMA Density	1892.363	48	48	85.1	63.2	1.17	0.86–1.60	0.32
Ki 67 Density	1097.034	48	48	64.4	85.1	1.30	1.06–1.61	0.01
Caspase Density	205.198	47	49	85.8	68.6	1.26	1.02–1.56	0.03
SMA-Ki67 nK(25)	1.167	46	48	69.6	68.6	0.98	0.71–1.34	0.90
SMA-Caspase nK(25)	1.106	38	44	85.1	67.6	0.92	0.65–1.30	0.64
Caspase-Ki67 nK(25)	1.163	42	42	67.6	89.7	0.82	0.58–1.16	0.26
**B.**								
**Metric**	**Cutoff**	**N High**	**N Low**	**Survival High (mo.)**	**Survival Low (mo.)**	**Cox HR**	**95% CI**	***p***
SMA H-score	36.346	10	60	32.6	47.6	1.51	1.05–2.17	0.03
Ki 67 H-score	7.06	27	43	33.3	56.5	1.47	1.13–1.91	<0.001
Caspase H-Score	1.137	13	57	46.7	46.7	1.19	0.89–1.58	0.24
SMA Density	1911.912	35	35	46.7	35.9	1.24	0.91–1.68	0.17
Ki 67 Density	1097.034	36	34	40.5	46.7	1.21	0.98–1.50	0.08
Caspase Density	189.285	35	35	86.1	40.5	1.18	0.95–1.46	0.13
SMA-Ki67 nK(25)	1.139	35	33	40.5	46.7	0.94	0.67–1.31	0.70
SMA-Caspase nK(25)	1.085	28	32	47.6	46.7	0.83	0.56–1.26	0.39
Caspase-Ki67 nK(25)	1.131	31	31	54.1	40.5	0.72	0.47–1.09	0.12
**C.**								
**Metric**	**Cutoff**	**N High**	**N Low**	**Survival High (mo.)**	**Survival Low (mo.)**	**Cox HR**	**95% CI**	***p***
SMA H-score	37.2	30	47	27.2	48.9	1.46	1.06–2.02	0.02
Ki 67 H-score	7.06	10	67	23.1	55.8	1.71	1.35–2.17	<0.001
Caspase H-Score	0.742	50	27	37.2	33.1	1.46	1.14–1.88	<0.001
SMA Density	1861.339	38	39	37.2	41.5	1.19	0.89–1.60	0.24
Ki 67 Density	1107.625	38	39	27.2	41.5	1.27	1.04–1.57	0.02
Caspase Density	174.07	37	40	36.4	37.2	1.25	1.02–1.53	0.03
SMA-Ki67 nK(25)	1.156	37	38	36.4	37.2	1.05	0.76–1.45	0.76
SMA-Caspase nK(25)	1.1	31	35	34.2	37.2	1.03	0.76–1.42	0.84
Caspase-Ki67 nK(25)	1.164	35	33	33.1	38.4	0.92	0.65–1.31	0.65

## Data Availability

The data that support the findings of this study are available from the corresponding author upon reasonable request.

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
