# Peer review of "Geospatial Cellular Distribution of Cancer-Associated Fibroblasts Significantly Impacts Clinical Outcomes in Metastatic Clear Cell Renal Cell Carcinoma"

_cancers, 2021, doi:10.3390/cancers13153743_

Round 1

Reviewer 1 Report

In this manuscript, the authors detected CAFs, proliferating cells, and apoptotic cells by markers αSMA, Ki-67, and caspase-3, respectively. They studied the geospatial relationships among CAFs, proliferating cells, and dying cells by calculating H-scores and cellular density. They drew a conclusion that CAFs are spatially clustered with and closer to proliferating tumor cells. Furthermore, they analyzed their association with survival outcomes. They found that patient samples with higher tumor cell proliferation had worse overall survival and were more likely to be resistant to systemic tyrosine-kinase-inhibiting targeted therapies.

Their work provided us a new insight into the relationship between CAF and proliferating tumor cells from the geospatial perspective. In addition, they performed a relatively systematical analysis of the geospatial proximity of tumor cells and CAFs on overall survival and resistance to targeted therapies by combining clinical data and tumor prognosis analysis. These results can further help us understand the role of CAFs in renal cell carcinoma and provide a reference for other similar carcinogenic mechanisms.

However, there are still some concerns that the authors should address.

Major concerns:

  1. αSMA is a marker for normal fibroblast but not specific for CAFs. For this study, a CAF-specific marker should be used.
  2. There are sub-classes of CAFs due to the heterogeneity of CAFs, the authors should discuss it or show the results of these subtypes of CAFs individually.
  3. Line 165: the cutoff standard needs further explanation. A normal kidney sample can be used as a control to define the cutoff.

Minor concerns:

  1. Please rephrase the sentence in line 276.
  2. For some statistical analysis methods, the authors could provide a brief introduction to help the reviewer/reader understand the meaning and purpose of the method.

Reviewer 2 Report

I like the paper and the content. It just needs minor corrections . Below are my suggestions.

The sentence in line 24-26 is too long, a run on sentence. You need to break it up. Maybe try saying this-

Cancer-associated fibroblasts are highly prevalent cells in the clear cell renal cell carcinoma (ccRCC) tumor immune microenvironment. They are thought to potentiate tumor proliferation primarily through paracrine interactions, as evidenced by laboratory-based studies.[ ALSO GIVE YOUR REFERENCE FOR THE “LABORATORY-BASED STUDIES” THAT YOU ARE REFFERING TO HERE]

Line 46-50- another run on sentence. Try to punctuate a bit more whenever possible. When a sentence is too long, it becomes too hard to follow .

The survival analyses used an optimal cut-point method, maximizing the log-rank statistic, to stratify the IHC-derived metrics into high and low groups. Then multivariable Cox regression analyses were performed accounting for age and International Metastatic RCC Database Consortium (IMDC) risk category.

Line 57 did you mean to say “regarding Ki-67” instead of “regarding CAF”. Because  in the very next line you state “For Caspase-3”… So it seems like in the 2 sentence you are comparing and contrasting results from Ki-67 versus  that  from Capsase-3???

Line 98- Here is another run on sentence, put a period after you stated your first aim, the state your next aim.

……….relationships between CAFs and proliferating and apoptotic tumor cells in primary tumor samples from patients with metastatic ccRCC. Additionally,  to evaluate the associations between these measures and OS.

Line 381- what you are saying is understandable, but be more specific. It would be good to say -    

……. with tumor cells further away from CAFs  succumbing to apoptosis;

Line 397- 400 is a very long, run -on sentence here again. Maybe put a period after “expression” in line 399, then start a new sentence.

 Similarly, hypoxia in the tumor microenvironment has been identified as a significant determinant of the extracellular matrix composition in tumors, resulting in increased HIF1A expression. This subsequently increases the production of growth factors that can potentiate tumor cell proliferation and trigger fibroblast activation and fibrosis [27]

Line 400-403 another run on sentence. Try not to have sentence loner than 2 lines without a period, so a four line sentence is far too long. You can lose the reader/your intended meaning with very long sentences.

In the 400 >80% of ccRCC tumors that harbor somatic VHL mutations, the resulting derangement of  HIFα metabolism would be expected to result in a tumor microenvironment consistent with extreme hypoxic conditions. This resultant state would be in place regardless of the true state of oxygen availability within the tissue.

Line 408-410. Please work on the run on sentences

This finding corroborates previous IHC studies of ccRCC patients that identified an association between increasing CAF density and OS, as well as more advanced stage at diagnosis.  However, these cohorts did not include stage IV patients [4-6].

Line 412  “While Using Ki-67 staining,…….”

Line 435

 There are several limitations of our study that deserve mentioning

Line 435- long sentence- see suggestion-

First, the patient cohort studied was heterogenous and included a variety of systemic treatment agents and was determined partially on the availability of tumor samples. This limited the applicability of our results with contemporary systematic treatment regimens.

Line 438- Secondly

Reviewer 3 Report

This is an interesting paper that addresses the question whether and to what extent the spatial distrubution of CAFs and proliferating/apoptotic RCC cells determines patient outcome. However, there are a number of deficiencies that preclude publication of the study in its present form.

  1. Representative microphotographs (low and high magification) of the stainings must be included. Ideally, a graphic overview of the analyses should also be shown.
  2. It is not clear to me what the authors mean with "geospatial", "spatial" should be sufficient.
  3. 10 specimens from normal kidney were used as controls, but what was the source? Was there an ethical approval?
  4. The authors use Ki-67 as a marker for proliferating cancer cells, but  Ki-67 is also overexpressed in proliferating immune cells for instance. How can the authors be certain that only cancer cells are stained? 
  5. α-Smooth muscle actin is expressed in normal (activated) fibroblasts or other cell types. More markers are needed to make sure that the positive cells are truly CAFs, at least in a number of tumors as proof-of-concept. 
  6. Using only three markers is an over-simplification of the complex biological processes that underlie tumor-CAF interactions. The authors need to tone down some conclusions.
  7. The authors claim that clusting of CAFs and tumor cells potentiates tumor cell proliferation (p. 15). What is the evidence for this claim. It could also be the other way i.e., tumor cells stimulate CAFs. 

Round 2

Reviewer 1 Report

The authors addressed my concerns. However, some sentences in the new version can be reworded to make them easier to be understood.

Author Response

Thank you for your feedback. We were not entirely sure which sentences were referred to, but we went through the entire manuscript and clarified any potentially confusing points to the best of our ability. Thank you again.  

Reviewer 3 Report

My points of critique have been sufficienlty addressed.

Author Response

Thank you for your feedback and for your thoughtful review.